# Network meta-analysis of the efficacy of endoscopic cardia peripheral tissue scar formation (ECSF) in the treatment of gastroesophageal reflux disease

Chaoyi Shi[1], Shunhai Zhou[1], Xuanran Chen[2], Diyun Shen[3], Tianyue Wang[3], GeSang ZhuoMa[4], Mingzhi Feng[4], Yan Sun[4], Jun Zhang[5]*

1 The Second Clinical Medical College, Zhejiang Chinese Medical University, Hangzhou, Zhejiang, China,
2 The Second School of Clinical Medicine, Hangzhou Normal University, Hangzhou, Zhejiang, China, 3 The First Clinical Medical College, Zhejiang Chinese Medical University, Hangzhou, Zhejiang, China,
4 Department of Gastroenterology, Zhejiang Provincial People's Hospital, (Affiliated People's Hospital, Hangzhou Medical College), Hangzhou, Zhejiang, China, 5 Department of Gastroenterology, The First Affiliated Hospital of Zhejiang Chinese Medical University (Zhejiang Provincial Hospital of Chinese Medicine), Hangzhou, Zhejiang

* zhangjun1@hmc.edu.cn

**Data Availability Statement:** All relevant data are within the manuscript and its Supporting Information files.

## Abstract

Endoscopic antireflux therapy is widely used in clinical practice. Peroral endoscopic cardial constriction (PECC), antireflux mucosal intervention (ARMI), and radiofrequency ablation (RF) possess analogous antireflux mechanisms. This comprehensive systematic review and meta-analysis aimed to evaluate and compare the safety and effectiveness of antireflux therapy during endoscopic cardia peripheral tissue scar formation (ECSF) procedures. We comprehensively searched the Web of Science, PubMed, Embase, China National Knowledge Infrastructure, and Wan-Fang databases for articles published from January 1990 to January 2024. Network meta-analysis (NMA) was used to assess the outcomes, with outcome metrics including the Gastroesophageal Reflux Questionnaire (GERD-Q) score, proton pump inhibitor discontinuation rate, pH <4.2 percent acid reflux time (AET), lower esophageal pressure (LES pressure), DeMeester score, adverse events, and patient satisfaction. Twenty studies involving 1219 patients were included. PECC was significantly superior to RF in lowering the patients' postoperative GERD-Q scores(MD = -2.34, 95% confidence interval (CI): [-3.02, -1.66]), augmentation of LES pressures(MD = 3.22, 95% CI: [1.21, 5.23]), and having a lower incidence of serious adverse events. ARMI was preferable to PECC (MD = -2.87, 95% CI [-4.23, -1.51])and RF (MD = -1.12, 95% CI [-1.79, -0.54]) in reducing the AET percentage, but was not as effective as PECC in lowering GERD-Q scores(MD = -1.50, 95% CI [-2.47, -0.53]). The incidence of adverse effects was less than 10% for all interventions, with most of them mildly self-resolving. Each ECSF procedure resulted in a favorable outcome in patients with GERD. Considering the safety and efficacy of treatment, PECC was the most favorable choice among ECSF procedures.

**Funding:** Supported by the Provincial and ministerial joint project, WKJ-ZJ-2018 (funding received by JZ), Clinical Helicobacter pylori adhesion subtype analysis and vaccine design and screening, and the Zhejiang Provincial Science and Technology Program of Traditional Chinese Medicine, 2023ZL266, Protective effect of isolicorice on ulcerative colitis in mice through NF-κB signaling pathway and its mechanism. The funders had no role in study design, data collection and analysis, decision to publish, or preparation of the manuscript.

**Competing interests:** The authors have declared that no competing interests exist.

## 1. Introduction

Gastroesophageal reflux disease (GERD) is a prevalent gastrointestinal disorder worldwide, with a global prevalence of 13.3%. The prevalence of GERD in China is 12.5% [1], and its incidence has been on the rise recently [2, 3]. An imbalance between the reflux and the esophageal antireflux anatomical barrier is the main cause of GERD pathogenesis [4]. Most patients experience reduced quality of life owing to symptoms such as heartburn, reflux, or dysphagia and their associated long-term complications. GERD significantly impacts the physical, mental, and social well-being and work efficiency of patients, posing a significant burden [5, 6].

The current treatment for GERD begins with lifestyle changes and pharmacological therapy. Proton pump inhibitors (PPIs) form the bedrock of pharmacological therapy for GERD [7]. PPIs administered for 4 weeks can achieve a remission rate of approximately 70% in patients with erosive esophagitis [8], with an expected cure rate of up to 90% after 8 weeks of administration [9]. However, nearly 40% of patients with reflux symptoms do not respond well to standard PPI therapy and may subsequently develop refractory GERD (RGERD) [3]. Moreover, long-term use of PPIs increases the risk of gastrointestinal infections, hip fractures, and kidney disease [3, 10]. Patients with RGERD should be evaluated for invasive antireflux surgery (ARS) when medications are not effective [11]. The most widely performed form of ARS is laparoscopic antireflux surgery (LARS). However, following the implementation of LARS, adverse effects, such as dysphagia and gas flatulence syndrome, arise. Owing to its increasingly severe damage and detrimental consequences, the use of LARS is diminishing annually [12].

Endoscopic minimally invasive antireflux treatment is continuously being explored. Peroral endoscopic cardial constriction (PECC), antireflux mucosal intervention (ARMI), and radiofrequency ablation (RF) are widely used in clinical practice, taking into consideration both safety and feasibility [13, 14]. Moreover, there are similarities in the antireflux mechanisms of RF, PECC, and ARMI. Through scar tissue formation at the gastroesophageal junction, these procedures achieve the common goal of antireflux by exerting greater pressure on the lower esophageal sphincter (LES) and its surrounding tissues, thereby enhancing the nonexternal force on the cardiac and adjacent tissues. Given their shared antireflux mechanism, the term "endoscopic cardia peripheral tissue scar formation" (ECSF) was introduced to collectively describe them. Previous studies [15] have demonstrated that all ECSF procedures have good efficacy in patients with RGERD; however, studies comparing the efficacy of the three procedures are still rarely reported. This study aimed to compare the efficacy of various ECSF procedures in patients with GERD.

## 2. Materials and methods

This meta-analysis was reported in accordance with the principles and requirements of PRISMA and pre-registered with PROSPERO under the registration number CRD42024519188.

### 2.1. Search strategy

We comprehensively searched for articles published from January 1990 to January 2024 using the China National Knowledge Infrastructure, Wanfang Database, PubMed, Embase, and Web of Science databases. The languages used were English and Chinese. The search keywords were as follows: ((ARMS) or (antireflux mucosectomy) or (ARMA) or (antireflux ablation) or (PECC) or (cardial constriction) or (radiofrequency) or (stretta) or (antireflux surgery) or (endoscopic treatment)) and ((GERD) or (gastroesophageal reflux disease) or (esophageal reflux)). References to relevant reviews and meta-analyses were reviewed to ensure that no relevant studies were omitted.

## 2.2. Inclusion and exclusion criteria

Literature screening was performed independently by two researchers, with a third researcher resolving any disputes. The inclusion criteria were stipulated as follows: (1) Study type: randomized controlled trials (RCTs) or cohort studies. (2) Study population: patients with GERD who underwent endoscopic treatment, with no limitations imposed on race, age, sex, or occupation. (3) Interventions: studies comparing two or more of the following treatments: ARMI, RF, PECC, or PPIs. (4) Control group: treated with PPIs. (5) Outcome metrics: Gastroesophageal Reflux Questionnaire (GERD-Q) score, PPI discontinuation rate, percentage of time with acid reflux at pH < 4.2 (AET), lower esophageal pressure (LES pressure), DeMeester score, adverse events, and patient satisfaction, including studies with one or more outcome metrics with complete data. Studies were excluded if they met the following exclusion criteria: (1) patients from special populations (obese, children, or patients with a history of esophageal or gastric surgery); (2) studies without complete information; and (3) case reports, animal experiments, preclinical studies, reviews, and meta-analyses. (4) Enrollment of fewer than 10 patients; (5) studies with a follow-up time of less than 1 month; (6) in cases where multiple studies reporting on the same patient cohort were found, only the study with the longest follow-up duration or largest sample size was considered for inclusion.

## 2.3. Data extraction

Initially, irrelevant literature was excluded by reviewing titles and abstracts and then further screened by reading the full text based on the inclusion and exclusion criteria. An Excel spreadsheet was used to record the following details: (1) Study information: authors' names, nationalities, and year of study publication. (2) Study characteristics: study type, sample size, age, sex, follow-up time, and treatment method. (3) Relevant outcome indicators: GERD-Q score, discontinuation rate of PPIs, AET, LES pressure, DeMeester score, adverse effects, patient satisfaction, and other indicators. If the data format of the study outcome indicators did not match the study requirements, they were converted using appropriate formulas. If a study included more than one intervention group, only the group relevant to this study was analyzed. If multiple intervention groups were simultaneously eligible, the groups were merged according to the Cochrane Handbook of Systematic Evaluation 5.1.0 methodology [16, 17]. Data formats that did not align with study requirements were converted using appropriate formulas.

## 2.4. Quality assessment

The overwhelming majority of studies incorporated into the analysis were RCTs. The included studies were rigorously evaluated using the Cochrane Collaboration's recommended risk of bias assessment tool, encompassing criteria such as randomization, blinding, allocation concealment, completeness of outcome data, selective reporting of results, and potential for other biases. Each article was categorized as "low risk," "unclear," and "high risk" according to the actual situation of the study [18]. Some of the included studies were cohort studies, and their quality was evaluated using the Newcastle-Ottawa scale (NOS). The NOS quality evaluation encompassed three key aspects: subject selection, comparability, and outcome or exposure, with a number of evaluation entries under each item and a total of 9 points out of a possible score. In the entry for the duration of follow-up, a follow-up period of greater than 1 year receives 1 point; otherwise, no points are awarded. In the data completeness category, a score of 1 is awarded if the loss-to-follow-up rate is less than 10%; otherwise, no score is given. A loss-to-follow-up rate of less than 10% receives 1 point; otherwise, no points are awarded.

## 2.5. Statistical analysis

Statistical analysis of the included literature was conducted using STATA 14.0, and categorical variables were described by the odds ratio (OR) with 95% confidence intervals (CIs). Continuous variables are presented as mean difference (MD) along with a 95%. In the case of a closed loop, the assumption of consistency was evaluated using a node-by-node splitting approach for each outcome. A p-value of <0.05 was considered statistically significant. When there was no closed loop, no consistency check was required. The Surface Under the Cumulative Ranking (SUCRA) was employed to analyze and rank the outcome indicators, and we leveraged funnel plots to evaluate the risk of publication bias and study effects, particularly when the outcome metrics encompassed over 10 studies.

## 3. Results

### 3.1. Study characteristics and methodological quality

Based on the keywords in English and Chinese, 17,346 documents were retrieved during the initial examination. After screening, 20 papers, with a total of 1,219 patients, were finally included [19–38]. Fig 1 illustrates the literature screening process. Treatment regimens were divided into four groups: control (PPI treatment), PECC, RF, and ARMI. The full text of each included document was read, and the first author, year of publication, demographic data, sample size, intervention, follow-up time, and outcome measures (GERD-Q score, PPI discontinuation rate, AET percentage, LES pressure, DeMeester score, adverse reactions, patient satisfaction, and other information) were extracted (Table 1).

The methodological quality of all the included studies is shown in Fig 2, Tables 2 and 3. Randomized sequence generation methods were reported in all included RCT studies except for studies by Lu [24], Kalapala [27] and He [32] et al. However, many trials did not provide

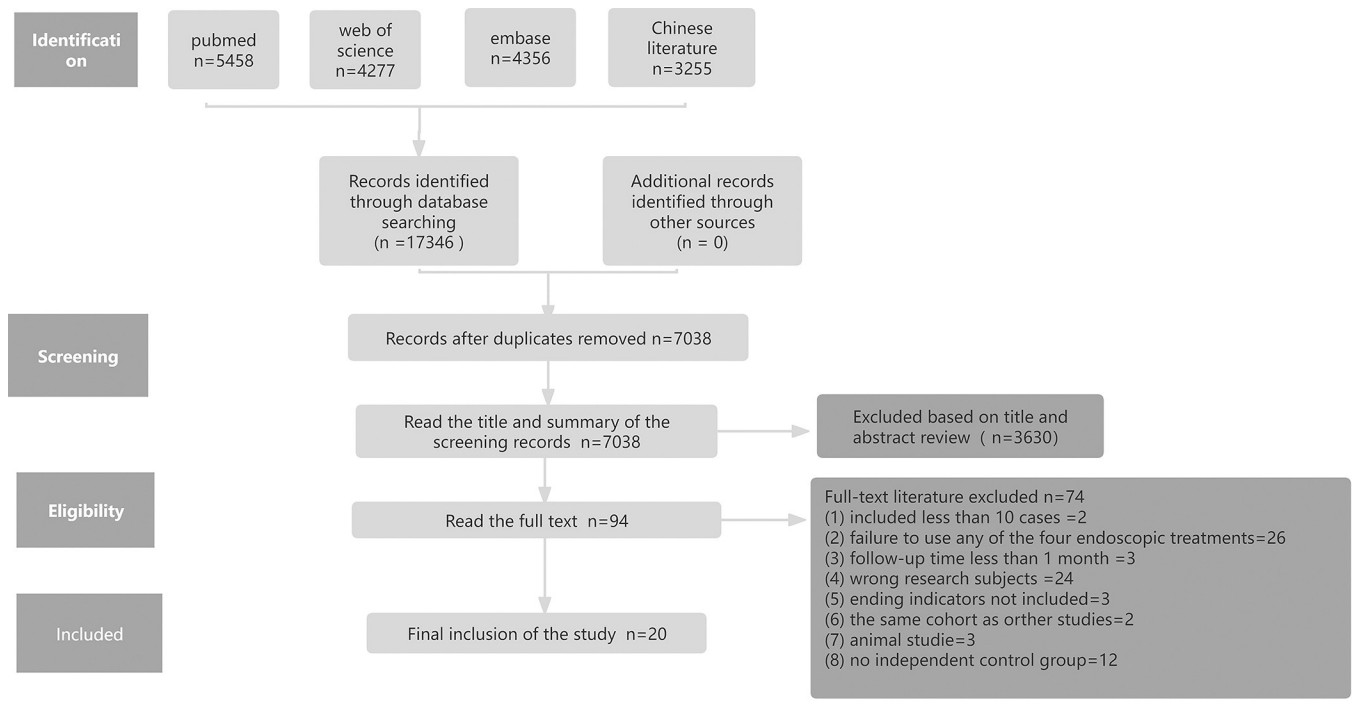

**Fig 1. Flow diagram showed the selection process of studies.**

**Table 1. Characteristics of included studie.**

| First author | Country | Patients | Publication years | Follow-up time | Arm1 | Arm2 | Design | Sex(M/F) | | Age(y) | | History of GERD(y) | | Outcome Measures |
|---|---|---|---|---|---|---|---|---|---|---|---|---|---|---|
| | | | | | | | | Arm1 | Arm2 | Arm1 | Arm2 | Arm1 | Arm2 | |
| LiJian Liu [19] | China | 30/30 | 2022 | 2months | PECC | RF | RCT | 11/19 | 13/17 | 51.13 ±3.82 | 49.56±4.83 | 3.25 ±1.21 | 3.67 ±1.26 | ①②⑤ |
| HuaShi [20] | China | 23/23 | 2021 | 6months | PECC | PPIs | RCT | 17/6 | 15/8 | 54.33±6.89 | 53.63±6.16 | 3.13±0.98 | 2.98 ±1.09 | ④⑤⑥ |
| HonggangLi [21] | China | 24/24 | 2022 | 2months | PECC | PPIs | RCT | 14/10 | 13/11 | 45.74 ±12.75 | 44.36 ±12.27 | 4.56±1.10 | 4.24 ±1.03 | |
| ShuaiTang [22] | China | 30/30 | 2020 | 1months | PECC | PPIs | RCT | 20/10 | 19/11 | 47.66 ±10.28 | 48.09 ±10.36 | 3.61±1.09 | 3.32 ±1.13 | ② |
| YanWang [23] | China | 18/16 | 2023 | 2years | ARMS | RF | RCT | 10/8 | 11/5 | 59.39 ±14.05 | 54.31 ±13.05 | 5.5±2.18 | 4.0±1.88 | ③④⑥ |
| Xin Lu [24] | China | 25/25 | 2019 | 12months | RF | PPIs | RCT | 19/6 | 17/8 | 53.3±5.6 | 50.8±6.2 | NA | NA | ③④⑥ |
| Waseem M [25] | Egypt | 75/75 | 2018 | 12months | PECC | PPIs | RCT | 49/26 | 50/25 | 39.3 ± 5.1 | 38.8 ± 4.8 | NA | NA | ④ |
| Ayman M [26] | USA | 12/12 | 2010 | 12months | RF | PPIs | RCT | 9/3 | 7/6 | 36.7 ± 9.5 | 32.0 ± 8.3 | 6.4 ± 5.3 | 5.2 ± 0.3 | ①②③④ |
| Rakesh Kalapala [27] | India | 10/10 | 2017 | 12months | RF | PPIs | RCT | 10/0 | 10/0 | 38.89 ±14.69 | 34.00 ±11.36 | NA | NA | ②③ |
| JianYi [28] | China | 22/22 | 2022 | 6months | PECC | PPIs | RCT | 14/8 | 13/9 | 43.25±4.17 | 43.22±4.14 | 4.21±0.46 | 4.19 ±0.48 | ③④⑥ |
| Yinping Wang [29] | China | 80/80 | 2022 | 12months | PECC | PPIs | RCT | 37/43 | 48/32 | 50.65 ± 2.46 | 51.47 ± 3.04 | 2.62±1.04 | 2.75 ±2.17 | ①③④⑤ |
| Jing Han [30] | China | 32/32 | 2021 | 3months | PECC | PPIs | RCT | 18/14 | 15/17 | 51.72 ±13.35 | 52.56 ±12.06 | 4.31±1.28 | 3.88 ±1.18 | ④ |
| J. Arts [31] | USA | 11/11 | 2012 | 3months | RF | PPIs | RCT | 5/17 | 5/17 | 46.5 ± 2.4 | | 5.1 ± 0.8 | | ②④ |
| Suyu He [32] | China | 28/21 | 2020 | 6months | RF | PPIs | RCT | 16/12 | 13/8 | 45.4±9.6 | 50.0±8.5 | 4.4±3.4 | 3.9±2.4 | ①②③④⑤ |
| Huixia Cao [33] | China | 34/30 | 2021 | 12months | PECC | ARMS | RCT | 18/16 | 17/13 | 56.29±9.99 | 57.03±8.83 | 4.84±1.96 | 4.29 ±1.46 | ③④ |
| YueChang [34] | China | 50/50 | 2020 | 12months | PECC | RF | Cohort study | 29/21 | 27/23 | 53.74±9.79 | 53.21±9.67 | 3.98±0.78 | 3.76 ±0.69 | ②④⑥ |
| Xinke Sui [35] | China | 39/30 | 2022 | 6months | ARMS | RF | Cohort study | 22/17 | 18/12 | 57.21 ±12.88 | 52.04±9.50 | NA | NA | ③④⑥ |
| Yanyan Zheng [36] | China | 28/20 | 2021 | 12months | PECC | ARMS | Cohort study | 12/16 | 13/7 | 56.32±8.19 | 51.90 ±10.11 | 4.0±2.89 | 5.5±4.81 | ①④⑤⑥ |
| Corley [37] | USA | 35/29 | 2003 | 12months | RF | PPIs | RCT | 16/19 | 17/12 | 45±12 | 52 ± 15 | NA | NA | ①②③④ |
| CORON [38] | France | 23/20 | 2008 | 6months | RF | PPIs | RCT | 16/7 | 14/6 | 50±10 | 47±14 | 10.0±10.5 | 9.5±6.0 | ①③④ |

Outcome Measures: ①AET,time to acid reflux at pH < 4.2;②lower esophageal pressure;③PPIs use;④adverse events;⑤DeMeester score;⑥GERD-Q score;patient satisfaction; studies without available data are not included in the table; RCT: randomized controlled trial; NA: not applicable; PECC: peroral endoscopic cardial constriction; ARMI: antireflux mucosal intervention; RF: radiofrequency ablation; PPIs: proton pump inhibitors.

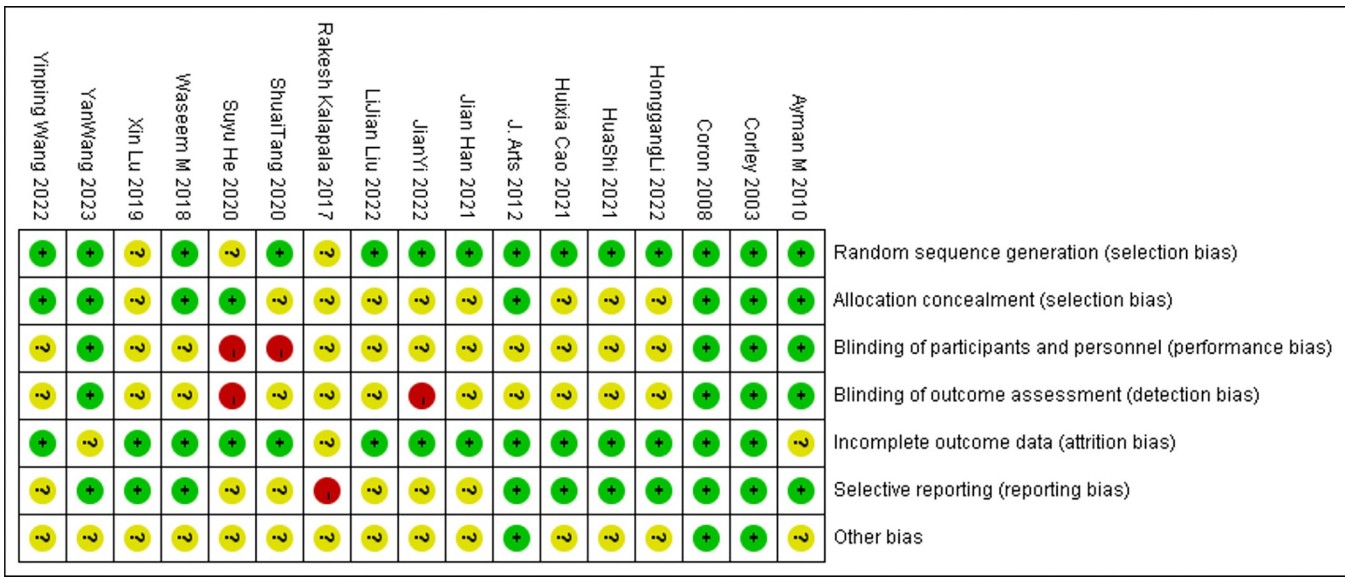

**Fig 2. Quality assessment of RCTs.**

details on blinding and allocation concealment, and more than 3/4 of the studies had an unclear bias towards blinded patients. Of the three cohort studies included, two scored 7, and one scored 6, all meeting the quality testing criteria.

## 3.2. Network meta-analysis of each outcome indicator

**3.2.1 GERD-Q scores.**   Seven papers reported changes in GERD-Q scores after ECSF. The evidence network presented in Fig 3A shows that the RF group has the largest node, representing the largest sample size, and the thickest line segment between the two nodes of RF and ARMI, representing the largest number of studies comparing RF with ARMI. The inconsistency test (p = 0.915) and node splitting method showed good consistency and no heterogeneity between studies (p>0.05). Accordingly, a consistency model was adopted for the analysis.

The SUCRA for each treatment was as follows: PECC (100.0%) > ARMI (64.8%) > RF (35.3%) > PPI (0.0%) (Fig 4A). The reticulated META analysis showed that all ECSF treatments reduced patients' GERD-Q scores compared to the PPI control group (PECC: MD = -5.30, 95% CI: -6.23, -4.36; RF: MD = -2.96, 95% CI: -4.02, -1.90; ARMI: MD = -3.80, 95% CI: -5.12, -2.48). Among ECSF procedures,PECC performed best. PECC significantly reduced GERD-Q scores compared to ARMI and RF (RF:MD = -1.50, 95% CI: -2.47, -0.53; RF:MD = -2.34, 95% CI: -3.02, -1.66) (Fig 5A).

**3.2.2 LES pressure.**   In total, 8 studies reported changes in LES pressure after ECSF, involving three interventions: PPI analogs, PECC, and RF. No relevant studies reported changes in LES pressure after ARMI. Fig 3B of the evidence network shows that the RF group has the largest node, representing the largest sample size, and that the thickest line segment

**Table 2. Quality assessment of cohort studies.**

| Study | Selection | Comparability | Outcome/ Exposure | Total Stars |
|-------|-----------|---------------|-------------------|-------------|
| Chang 2002 | 3 | 1 | 3 | 7 |
| Sui 2022 | 2 | 2 | 3 | 7 |
| Zheng 2021 | 2 | 1 | 3 | 6 |

**Table 3. SUCRA values and mean rank for different treatments for each outcome.**

| Treatment | SUCRA/% | PrBest/% | MeanRank |
|---|---|---|---|
| **GERD-Q scores** | | | |
| PECC | 100.0 | 99.9 | 1.0 |
| RF | 35.3 | 0.0 | 2.9 |
| ARMI | 64.8 | 0.2 | 2.1 |
| PPIs | 0.0 | 0.0 | 4.0 |
| **LES pressure** | | | |
| PECC | 100.0 | 99.9 | 1.0 |
| RF | 50.0 | 0.1 | 2.0 |
| PPIs | 0.1 | 0.0 | 4.0 |
| **AET** | | | |
| PECC | 66.6 | 0.2 | 2.0 |
| RF | 50.0 | 0.0 | 3.1 |
| ARMI | 100.0 | 99.9 | 1.0 |
| PPIs | 2.9 | 0.0 | 3.9 |
| **DeMeester scores** | | | |
| PECC | 74.6 | 36.9 | 1.8 |
| RF | 36.6 | 11.2 | 2.9 |
| ARMI | 70.7 | 50.5 | 1.9 |
| PPIs | 18.0 | 1.5 | 3.5 |
| **PPIs withdrawal rate** | | | |
| PECC | 75.1 | 50.1 | 1.7 |
| RF | 50.4 | 11.1 | 2.5 |
| ARMI | 73.8 | 38.8 | 1.8 |
| PPIs | 0.6 | 0.1 | 4.0 |
| **Patient satisfaction rate** | | | |
| PECC | 45.9 | 9.6 | 2.6 |
| RF | 79.7 | 54.5 | 1.6 |
| ARMI | 73.9 | 35.9 | 1.8 |
| PPIs | 0.5 | 0.0 | 4.0 |

SUCRA: surface under the cumulative ranking; PECC: peroral endoscopic cardial constriction; ARMI: antireflux mucosal intervention; RF: radiofrequency ablation; PPIs: proton pump inhibitors.

between the PPI control and RF nodes represents the largest number of studies in the included literature comparing PPIs to RF. The inconsistency test (p = 0.098) and node splitting method showed good agreement and no heterogeneity between the studies (p>0.05). Therefore, a consistency model was used for the analysis.

The SUCRA for each treatment was PECC (100.0%) > RF (50.0%) > PPI (0.1%) (Fig 4B). In addition, the reticulated META analysis revealed that both PECC and RF significantly increased the patients' postoperative LES pressure compared to the PPI control group (PECC: MD = 5.81, 95% CI: 3.61, 8.01; RF: MD = 2.59, 95% CI: 0.91, 4.27). Moreover, PECC was superior to RF, with a statistically significant difference (MD = 3.22, 95% CI: 1.21, 5.23) (Fig 5B).

**3.2.3 AET.** Seven studies reported changes in AET after ECSF. The evidence network in Fig 3C shows that the PECC group had the largest nodes, with the thickest line segment between the nodes of the PPI control and RF groups. The inconsistency test (p = 0.283) and node splitting method showed good consistency and no heterogeneity between studies (p>0.05). Therefore, a consistency model was used for the analysis.

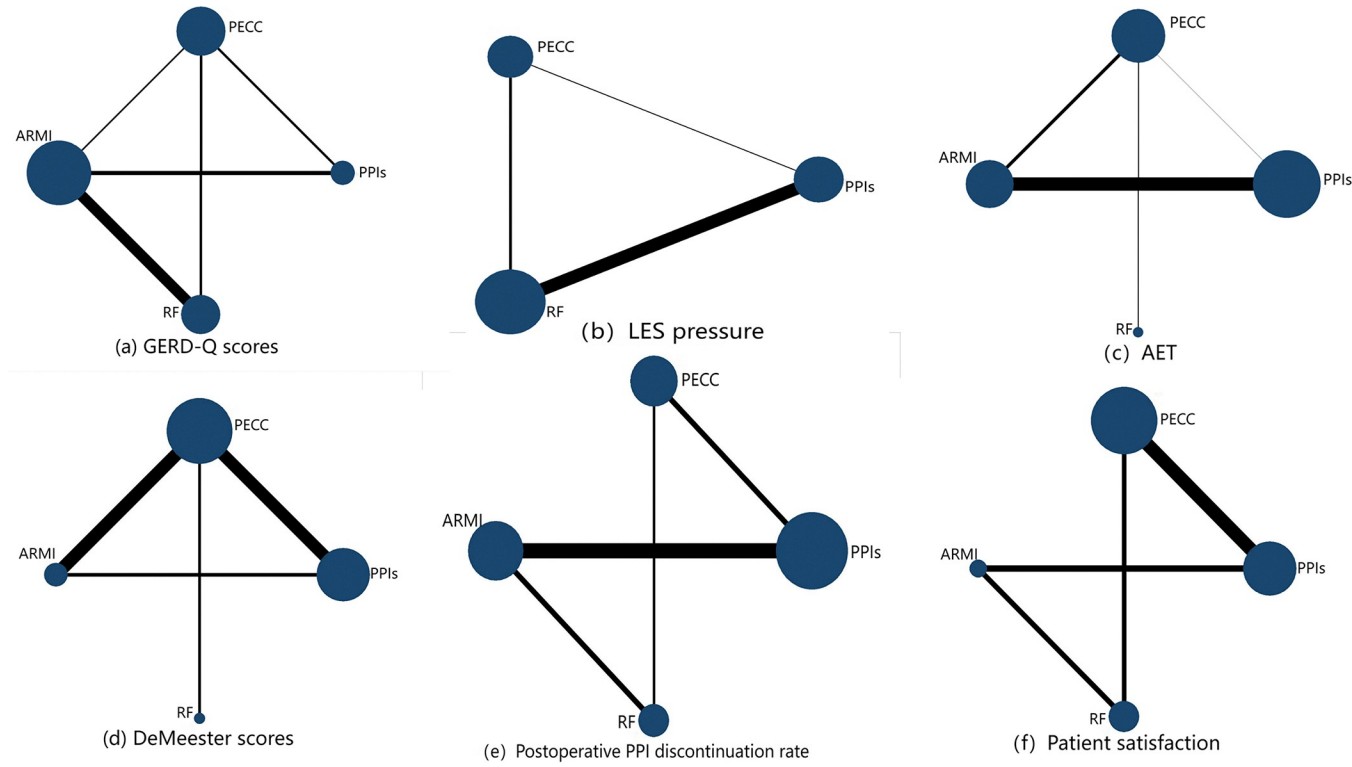

**Fig 3. Network of intervention comparisons.**

The SUCRA for each treatment was ARMI (100.0%) > PECC (66.6%) > RF (50.0%) > PPI (2.9%) (Fig 4C). The reticulated META analysis showed that both ARMI and PECC reduced the percentage of AET in postoperative patients compared with the PPI control group (ARMI: MD = -3.67, 95% CI: -4.42, -2.92; PECC: MD = -2.55, 95% CI: -2.87, -2.22). Compared with RF, the percentage reduction of AET by ARMI versus PECC was statistically significant (ARMI: MD = -2.87, 95% CI: -4.23, -1.51; PECC: MD = -1.75, 95% CI: -2.94, -0.56). Furthermore, ARMI showed superior performance to PECC (MD = -1.12, 95% CI: -1.79, -0.54) (Fig 5C).

**3.2.4 DeMeester scores.** Five of the included studies reported changes in DeMeester scores after ECSF. The evidence network in Fig 3D shows that the PECC group had the largest sample size and number of studies comparing the PPI and PECC groups. The inconsistency test (p = 0.845) and node splitting method showed good consistency and no heterogeneity between studies (p>0.05).

The SUCRA for each treatment measure was PECC (74.6.0%), ARMI (70.7%), RF (36.3%), and PPI (18.0%) (Fig 4D). However, the NMA outcomes revealed no statistically significant differences between the procedures (Fig 5D).

**3.2.5 Postoperative PPI discontinuation rate.** In total, 10 studies reported changes in PPI administration after ECSF. The evidence network in Fig 3E shows that the PPI control group has the largest node and thickest line segment between the nodes for the PPI control and RF groups, representing the largest sample size and number of studies, respectively, when comparing both groups. The inconsistency test (p = 0.468) and the node splitting method showed good consistency and no heterogeneity between the studies (p>0.05). Therefore, a consistency model was used for the analysis.

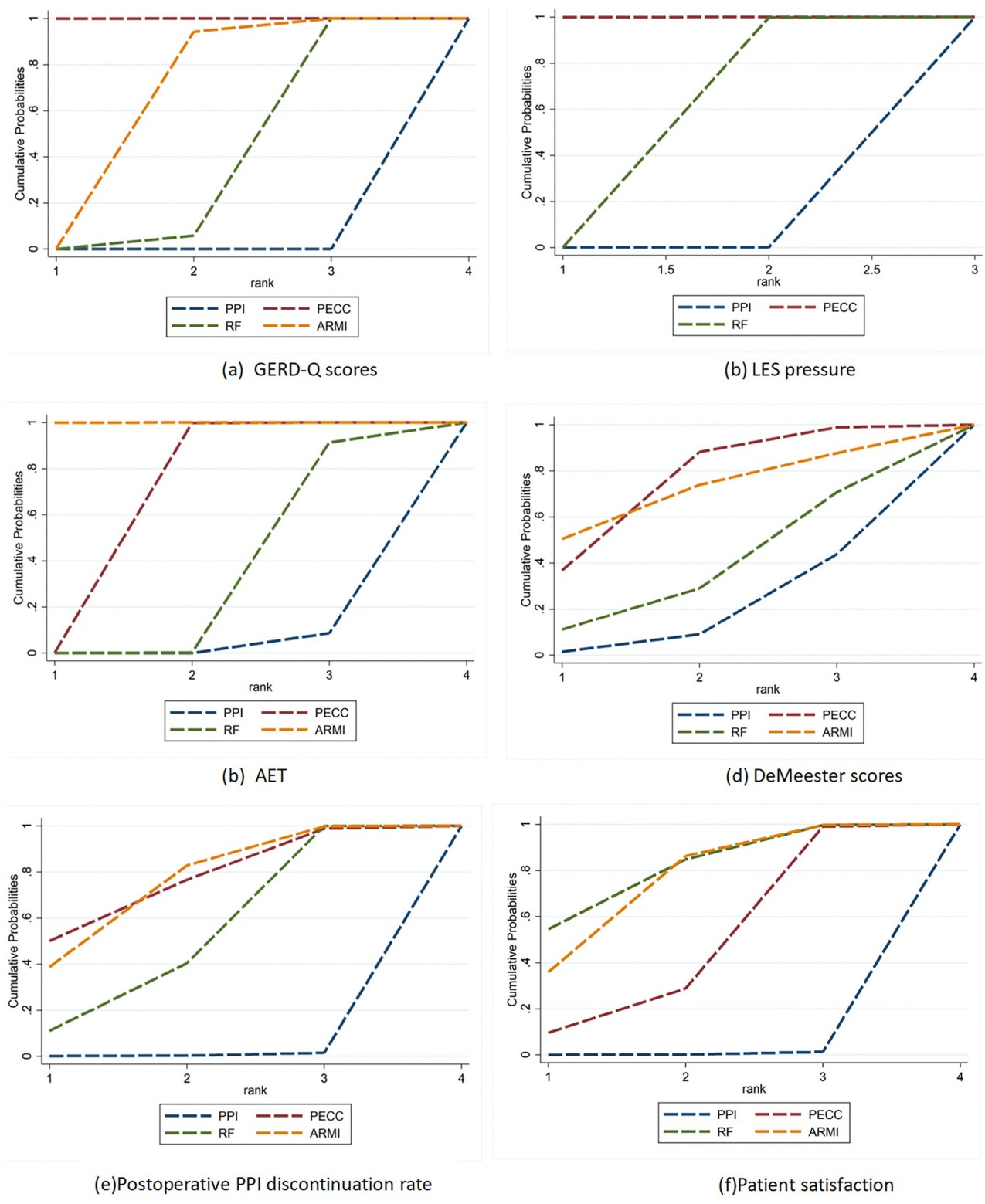

**Fig 4. SUCRA of intervention comparison.**

| PECC | 1.49 (0.52,2.46)* | 2.32 (1.64,3.00)* | 5.16 (4.27,6.04)* |
|---|---|---|---|
| -1.50 (-2.47,-0.53)* | ARMI | 0.83 (-0.21,1.87) | 3.66 (2.38,4.95)* |
| -2.34 (-3.02,-1.66)* | -0.84 (-1.88,0.20) | RF | 2.84 (1.81,3.86)* |
| -5.30 (-6.23,-4.36)* | -3.80 (-5.12,-2.48)* | -2.96 (-4.02,-1.90)* | PPI |

(a) GERD-Q scores

| PECC | -3.22 (-5.23,-1.21)* | -5.81 (-8.01,-3.61)* |
|---|---|---|
| 3.22 (1.21,5.23)* | RF | -2.59 (-4.27,-0.91)* |
| 5.81 (3.61,8.01)* | 2.59 (0.91,4.27)* | PPI |

(b) LES pressure

| ARMI | 1.12 (0.45,1.79)* | 2.87 (1.51,4.23)* | 3.67 (2.92,4.42)* |
|---|---|---|---|
| -1.12 (-1.79,-0.45)* | PECC | 1.75 (0.56,2.94)* | 2.55 (2.22,2.87)* |
| -2.87 (-4.23,-1.51)* | -2.87 (-4.23,-1.51)* | RF | 0.80 (-0.36,1.95) |
| -3.67 (-4.42,-2.92)* | -2.55 (-2.87,-2.22)* | -0.80 (-1.95,0.36) | PPI |

(c) AET

| PECC | -4.58 (-82.09,72.94) | 28.45 (-35.11,92.01) | 43.67 (-7.27,94.62) |
|---|---|---|---|
| 4.58 (-72.94,82.09) | ARMI | 33.03 (-67.14,133.19) | 48.25 (-44.34,140.84) |
| -28.45 (-92.01,35.11) | -33.03 (-133.19,67.14) | RF | 15.22 (-46.68,77.12) |
| -43.67 (-94.62,7.27) | -48.25 (-140.84,44.34) | -15.22 (-77.12,46.68) | PPI |

(d) DeMeester scores

| PECC | 0.95 (0.31,2.92) | 0.70 (0.22,2.25) | 0.16 (0.03,0.83)* |
|---|---|---|---|
| 1.06 (0.34,3.26) | ARMI | 0.74 (0.32,1.71) | 0.17 (0.05,0.58)* |
| 1.43 (0.44,4.59) | 1.35 (0.59,3.11) | RF | 0.23 (0.08,0.64)* |
| 6.28 (1.21,32.57)* | 5.93 (1.73,20.35)* | 4.40 (1.56,12.39)* | PPI |

（e）Postoperative PPI discontinuation rate

| RF | 0.88 (0.26,2.99) | 0.52 (0.11,2.34) | 0.11 (0.02,0.52)* |
|---|---|---|---|
| 1.13 (0.33,3.82) | ARMI | 0.58 (0.17,1.96) | 0.13 (0.03,0.58)* |
| 1.94 (0.43,8.77) | 1.71 (0.51,5.76) | PECC | 0.22 (0.06,0.75)* |
| 8.97 (1.91,42.23)* | 7.94 (1.74,36.23)* | 4.63 (1.34,16.05) * | PPI |

（f）Patient satisfaction

**Fig 5. The NMA results for different treatments.**

The SUCRA for each treatment were PECC (75.1%), ARMI (73.8%), RF (50.4%), and PPI (0.6%) (Fig 4E). The reticulated META analysis showed that all three ECSF procedures significantly increased the rate of discontinuation of PPI analogs and decreased their use compared with PPI controls (PECC: OR = 6.28, 95% CI: 1.21, 32.57; ARMI: OR = 5.93, 95% CI: 1.73, 20.35; RF: OR = 4.40, 95% CI: 1.56, 12.39). When the three procedures were compared, the differences were not statistically significant (Fig 5E).

**3.2.6 Patient satisfaction.** Six studies reported on patient satisfaction after ECSF. The evidence network in Fig 3F shows that the PECC group had the largest sample size and number of studies when comparing the PPI control and PECC groups. Analysis was performed using a consistency model (p = 0.957).

The SUCRA for each treatment was RF (79.7%) > ARMI (73.9%) > PECC (45.9%) > PPI (0.5%) (Fig 4F). The reticulated META analysis showed that patient satisfaction was higher for all three procedures in the ECSF group than in the PPI control group (RF: OR = 8.97, 95% CI: 1.91, 42.23; ARMI: OR = 7.94, 95% CI: 1.74, 36.23; RF: OR = 4.63, 95% CI: 1.34, 16.05). When the three procedures were compared, the differences were not statistically significant (Fig 5F).

**3.2.7 Adverse event.** Among the included studies, 15 reported adverse events after ESCF, accounting for a total of 14 serious adverse events (SAEs). SAEs occurred in two cases (0.47%) after PECC, nine cases (3.33%) after RF, and three cases (2.80%) after ARMI. The incidence of SAEs after PECC was lower than that for the other two procedures. The vast majority of SAEs that occurred in the study were gastrointestinal foramina and esophageal strictures, which were effectively controlled after re-endoscopic treatment and balloon dilatation. In addition, the main postoperative adverse events after ECSF included abdominal pain, chest pain, fever, and difficulty eating, which were mostly mild and resolved over time.

## 3.3. Publication bias

Publication bias was assessed for outcome metrics that included more than 10 papers, and only postoperative PPI discontinuation rates met this criterion. Fig 6 shows that the funnel plot had poor left-right symmetry and one dot scattered outside the 95% CI region, suggesting potential publication bias or a small sample effect in the included studies.

## 4. Discussion

According to the Rome Foundation, the Lyon Consensus, and the Montreal Consensus, GERD is a comprehensive disease with complex pathophysiologic mechanisms. Many patients with GERD have atypical clinical manifestations, unique predisposing factors, and different underlying causes, making a "one-size-fits-all" treatment unsuitable for all patients [39]. Currently, the goals of GERD treatment are to promote mucosal healing and control reflux symptoms [40] rather than "restore" or "avoid reflux".

The rapid development of endoscopic technology has led to its widespread use in GERD treatment. Numerous studies have examined the efficacy of various endoscopic treatments, but most focus on a specific procedure, lacking systematic comparisons of the efficacy of various endoscopic treatments. Previously, our team identified similarities in the antireflux mechanisms of PECC, RF, and ARMI, which led to the development of the concept of ECSF. We conducted a systematic review and meta-analysis of the efficacy and safety of ECSF [15] and found that all ECSF procedures were significantly effective and safe for patients with RGERD and had a much lower rate of SAEs compared to surgical treatment [41] and other endoscopic procedures [42]. However, the effectiveness of each ECSF procedure for patients with reflux and the identification of an optimal choice remain unclear. Therefore, this study systematically evaluated the efficacy of each ECSF procedure for patients with GERD using evidence-based

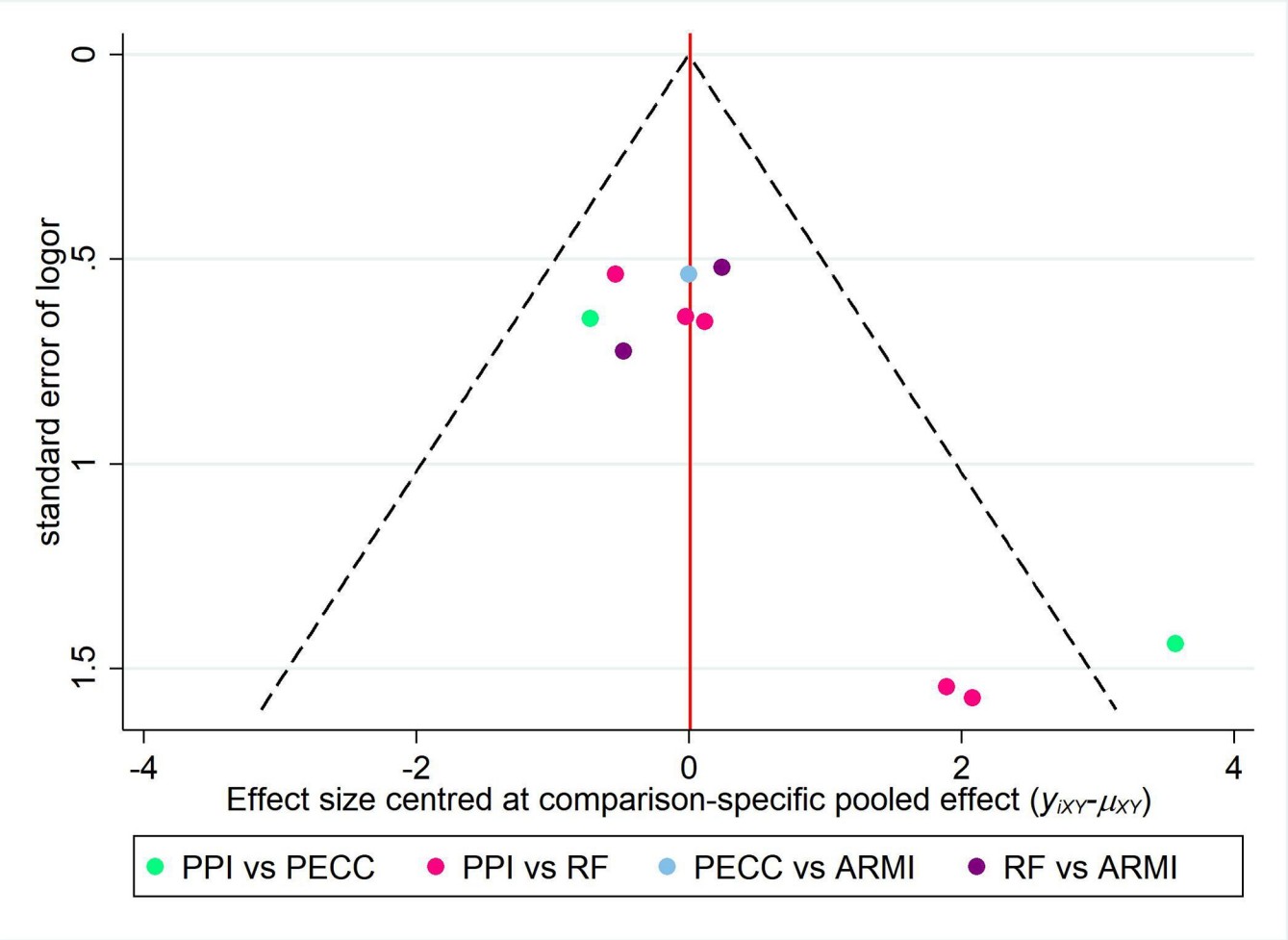

**Fig 6. The funnel plot for the included studies.**

medicine and compared their efficacy to provide evidence-based guidelines for clinical treatments and decision-making.

The GERD-Q score is a common clinical tool used for assessing reflux symptoms [43]. 24-hour oesophageal pH monitoring and esophageal manometry are commonly used to clinically assess the efficiency of subjective and objective indicators [44, 45]. Our results suggest that PECC is the most effective ESCF procedure based on GERD-Q scores before and after procedures. The NMA and probability ranking plot indicate that PECC has superior effects, while RF and ARMI are likely equally effective, with both being more effective than PPI. Similarly, PECC showed advantages over RF and PPI therapy in improving postoperative LES pressure and had the highest postoperative PPI discontinuation rate. Nevertheless, ARMI showed superiority over RF and PECC in improving oesophageal acid exposure. And RF perhaps results in higher patient satisfaction after treatment. Owing to limited data, we did not perform an NMA of adverse events after ECSF treatment. However, Table 4 shows that PECC had the lowest incidence of postoperative SAEs (0.47%) and a widely recognized safety profile, whereas the incidence of postoperative SAEs after RF and ARMI was similar to that previously reported after transoral incisionless fundoplication (TIF) [46].

**Table 4. Adverse events for different comparisons.**

| Study | SAEs | Adverse event | | |
| --- | --- | --- | --- | --- |
| | | PECC | RF | ARMI |
| Hua 2021 | 0 | 2Mild dysphagia,1Soreness | NA | NA |
| Wang 2022 | 0 | 1Bleeding, 1Mucosal damage | NA | NA |
| Lu 2019 | 0 | NA | 2Bloating, 1Bleeding | NA |
| Waseem 2018 | 0 | 19Mild dysphagia, 30Soreness | NA | NA |
| Ayman 2010 | 1 (RF: Pneumonia, pleural effusion) | NA | 0 | NA |
| Yi 2022 | 0 | 2Soreness,1Mild dysphagia | NA | NA |
| Han 2021 | 0 | 0 | NA | NA |
| He 2020 | 0 | NA | 2Soreness,1Infection,1Bloating | NA |
| Wang 2023 | 0 | NA | 5Soreness,1Bloating | 9 Stomach pain, 2Bloating, 5 Mild dysphagia |
| Cao 2021 | 1 (ARMS:Perforation)+2 (ARMS:Esophageal stenosis) | 2Mild dysphagia,3Infection, 3Soreness | NA | 3 Infection, 4 Soreness |
| Chang 2020 | 8 (RF:3 Perforation+5 Esophageal stenosis)/2 (PECC:2Esophageal stenosis) | 2Bleeding,1Infection | 5 Bleeding, 4 Infection | NA |
| Sui 2022 | 0 | NA | 0 | 1 Mild dysphagia |
| Corley 2003 | 0 | NA | 1 Bleeding | NA |
| CORON 2008 | 0 | NA | 5 Soreness, 2 Infection | NA |
| Zheng 2021 | 0 | 0 | NA | 0 |

SAEs:serious adverse events; PECC: peroral endoscopic cardial constriction; ARMI: antireflux mucosal intervention; RF: radiofrequency ablation; PPIs: proton pump inhibitors.

PECC has the lowest incidence of postoperative SAEs and has a higher safety profile than other therapeutic measures, mainly due to its treatment modality. PECC is a therapeutic procedure proposed by Professor Linghu in 2013 [47]. PECC is mainly performed by ligating the mucosa and part of the muscular layer at the gastro-oesophageal junction and releasing the ligature ring for ligation. Scarring is formed through constant contraction, thus forming an anti-reflux barrier for the treatment of gastro-oesophageal reflux. Compared with other procedures, PECC has the advantages of shorter procedure time and less invasiveness, as well as its efficacy has been well documented in numerous previous studies [48, 49]. In patients without severe reflux oesophagitis (LA class C or D) or oesophageal hiatal hernia greater than 2 cm, PECC is the preferred endoscopic treatment option, with fewer postoperative complications and lower medical costs [50].

Antireflux mucosal intervention (ARMI) includes anti-reflux mucosectomy (ARMS) and anti-reflux mucosal ablation (ARMA) [14]. ARMI is mainly performed by removing or

ablating part of the mucosa at the gastro-oesophageal junction, and scarring is formed by tissue contracture during the repair process, which serves to prevent reflux from entering the oesophagus. The present study showed that ARMI showed better results in improving oesophageal acid exposure. Previously, researchers have statistically analysed endoscopic treatments, including ARMS, RF, and transoral incisionless fundoplication (TIF), showing that ARMS is stronger than RF and TIF in reducing patients' acid reflux time, decreasing the use of acid-suppressing medications postoperatively, and improving patients' quality of life [51]. This is consistent with the results of our study. However, there exists a small probability of dysphagia due to oesophageal stricture after ARMI, and may require further dilation [52]. In clinical practice, ARMI pro-cedures are effective and safe for both PPI-refractory and PPI-dependent patients [14]. Patients with extra-oesophageal GERD who have mild to moderate morphological damage to the gastro-oesophageal junction are the most desirable target population for ARMI [53].

RF is mainly applied to the intrinsic muscular layer of the gastro-oesophageal junction using the Stretta system. It uses the thermal effect of radiofrequency to ablate the tissue, postoperative fibrosis of the lamina propria to form a scar, and reinforcement of the LES anti-reflux barrier. The advantage of RF over other treatments is that it reduces the frequency of transient relaxation of the LES. If the patient's valve morphology is normal when observed endoscopically, but esophageal dynamics show an increased frequency of transient relaxation of the LES, then RF may be a more appropriate treatment modality for these patients. However, a number of previous studies have questioned the efficacy of RF [54], and some guidelines are not very favourable to the Stretta procedure [55]. This study shows that RF, although it may not be as effective as the other two in improving reflux symptoms and oesophageal acid exposure, it may be supported by a higher level of patient satisfaction. The reason for this may be that RF is well tolerated by patients and that RF can exist as a post-surgical remedy. RF can be used in patients with failed LARS and can be used repeatedly as a remedial treatment after failed surgical procedures [56]. In addition, RF is an option for patients with mild oesophagitis who do not have Barrett's oesophagus [57].Similarly, there may be a role for RF in patients with reflux hypersensitivity or functional heartburn [58].

Several meta-analyses have evaluated the efficacy of different endoscopic treatments [51, 59]; however, they often include monolithic or fundamentally different interventions, reducing their clinical guidance value. This study is the first to simultaneously statistically analyse PECC, ARMI, and RF, which possess similar anti-reflux principles, to provide more data to support the choice of treatment for GERD. In the future clinical diagnosis and treatment of GERD, clinicians should recommend patients to complete the relevant questionnaires and various examinations. According to the results, the degree of reflux and the presence of morphological damage will be assessed, so that the most appropriate treatment can be chosen from the patient's actual situation.

This study had the following limitations: (i) Although most of the included studies were RCTs, some had small sample sizes, and the consistency test indicates minor variations and biases, highlighting the need for further large-scale, double-blinded, multicenter studies; (ii) The included RCT studies mostly had unknown risk in terms of allocation concealment, blinding, and other literature quality assessment, which may have potential risk of bias and ultimately affect the study's authenticity; (iii) The follow-up time varied greatly, with some studies having a shorter follow-up period and insufficient long-term follow-up data, necessitating a longer follow-up period and data collection to validate the long-term efficacy; (iv) The amount of literature on certain treatment modalities is limited. For example, only four articles on ARMI, involving 107 patients, were included, which is fewer than other ECSF treatments, affecting the credibility of the results; (v) Differences exist in the selection of research

participants among different studies. The vast majority of studies included participants with RGERD, although their diagnostic criteria for RGERD were outdated compared to the latest Lyon Consensus 2.0 [40]. However, a few studies [19, 34] did not explicitly emphasize the diagnosis of RGERD in the included patients. He et al. emphasized that the enrolled patients were clearly diagnosed with non-erosive reflux disease. This may have contributed to the heterogeneity; (vi) PECC is currently widely practised in China, but has not gained large-scale popularity in other regions. Most of the studies on PECC treatment inclusion were conducted in Chinese research institutes, and the results may be regionally biased. Similarly, most of the study populations included in this study were of Asian origin and ethnically biased. Conclusions are more appropriate for Asian populations.

## 5. Conclusion

This study initially compared the clinical efficacy of three ECSF procedures in patients with GERD. The results showed that PECC showed optimal outcomes in terms of improving postoperative GERD-Q scores, increasing LES pressure, and PPI discontinuation rate, with the lowest incidence of postoperative SAEs. ARMI was the most effective in improving esophageal acid exposure. Considering overall efficacy and safety, PECC is the most preferred ECSF procedure; however, selecting the most appropriate treatment should be based on the patient's specific situation.

## Supporting information

**S1 Checklist. PRISMA 2020 checklist.**
(DOCX)

**S1 Table. The data of meta-analysis.**
(XLSX)

**S2 Table. List of excluded studies.**
(XLSX)

**S3 Table. Risk of bias assessment.**
(XLSX)

## Author Contributions

**Conceptualization:** Chaoyi Shi, Jun Zhang.

**Data curation:** Chaoyi Shi.

**Formal analysis:** Chaoyi Shi, Shunhai Zhou, Mingzhi Feng.

**Investigation:** Tianyue Wang.

**Methodology:** Xuanran Chen, Diyun Shen.

**Project administration:** Yan Sun.

**Resources:** Jun Zhang.

**Supervision:** Tianyue Wang, GeSang ZhuoMa.

**Validation:** Diyun Shen, GeSang ZhuoMa.

**Visualization:** Shunhai Zhou, Xuanran Chen.

**Writing – original draft:** Chaoyi Shi, Jun Zhang.

**Writing – review & editing:** Jun Zhang.

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
