## [Decision Letter · Decision Letter 0]

30 Aug 2024

PONE-D-24-27892Network meta-analysis of the efficacy of endoscopic cardia peripheral tissue scar formation (ECSF) in the treatment of gastroesophageal reflux diseasePLOS ONE

Dear Dr. Zhang,

Thank you for submitting your manuscript to PLOS ONE. After careful consideration, we feel that it has merit but does not fully meet PLOS ONE’s publication criteria as it currently stands. Therefore, we invite you to submit a revised version of the manuscript that addresses the points raised during the review process.

We look forward to receiving your revised manuscript.

Kind regards,

Academic Editor

PLOS ONE

Journal Requirements:

Additional Editor Comments:

Please follow Reviewers' comments to update this manuscript. In addition, please increase the font size of A, B, C, D in Fig.3 Network of intervention comparisons. The current font size is too small to read.

Reviewers' comments:

Reviewer's Responses to Questions

**Comments to the Author**

1. Is the manuscript technically sound, and do the data support the conclusions?

Reviewer #1: Yes

Reviewer #2: Yes

2. Has the statistical analysis been performed appropriately and rigorously? 

Reviewer #1: Yes

Reviewer #2: Yes

3. Have the authors made all data underlying the findings in their manuscript fully available?

Reviewer #1: Yes

Reviewer #2: Yes

4. Is the manuscript presented in an intelligible fashion and written in standard English?

Reviewer #1: Yes

Reviewer #2: Yes

5. Review Comments to the Author

Reviewer #1: This comprehensive systematic review and meta-analysis evaluates and compares the safety and effectiveness of antireflux therapies during endoscopic cardia peripheral tissue scar formation (ECSF) procedures. The study finds that each ECSF procedure demonstrates favorable outcomes for patients with GERD. Among these procedures, PECC is identified as the most preferred option due to its superior safety profile and efficacy.

This is an interesting topic that worth of in-depth research. The paper offers some interesting results under the framework of network meta-analysis. I still have some questions to help improve the quality of the paper

1. In the introduction, the authors listed several current treatments for GERD, including PPIs, ARS and ECSF. Although they claimed that “the most widely performed form of ARS is laparoscopic antireflux surgery (LAS) ”, they focused solely on comparing the efficacy of various ECSF procedures with PPIs in patients with GERD. The rationale for excluding LAS from this comparison could be elaborated to provide further insights into their study design.

2. The colors of the scatter plots in “Fig.6 The funnel plot for the included studies” should exhibit greater contrast, as the current colors make it difficult to distinguish between points.

3. The manuscript does not clearly address why authors have chosen their outcome metrics. More literature about it should be cited.

4. In the discussion, the authors claimed, “However, the effectiveness of each ECSF procedure for patients with reflux and the identification of an optimal choice remain unclear.” Consequently, the manuscript should address how current clinical treatment decisions are made within each ECSF procedure. Additionally, it should explore how the conclusions drawn from the manuscript can be applied to clinical decision-making.

Reviewer #2: General Comments: The authors should be commended for providing a detailed and clearly-written summary of the network meta analysis. In particular, all methods are clearly described, relevant data are (abundantly) provided, and conclusions do not extend beyond what the results show. I have some minor concerns (below), but the reported Figures need better explanation (in terms of axis titles, legends or footnotes) in order to be helpful to readers.

Specific Comments:

1. Abstract: The authors report PECC is superior to RF with 2 numerical results (MD = -2.34; MD = 3.22), but then list three outcomes (GERD-Q score, LES pressure, and Serious adverse events). As readers will not be familiar with their approaches, it will not be obvious that the reported MD values apply to the first 2 outcomes. Please revise and clarify.

2. Abstract: In the next sentence, the authors report that ARMI was superior to "the other two," without specifying which "two" groups they're referring too, nor is it clear which outcomes the two reported results refer to. Please revise and clarify.

3. Methods Section: Great job on providing a detailed explanation of methods.

4. Figures 3-5: These figures will be practically uninformative to all but the most statistically-trained readers. Please include (as appropriate) axis titles, legends, or footnotes to explain what's included in this Figures. In particular, where there are several panels in a Figure, please indicate which outcome or group each refers to. In Figure 4, please consider an alternative schema, as the dashed lines are difficult to distinguish.

6. PLOS authors have the option to publish the peer review history of their article (what does this mean?). If published, this will include your full peer review and any attached files.

Reviewer #1: **Yes: **Yizhuo Chang

Reviewer #2: No

---

## [Author Response · Author response to Decision Letter 0]

12 Sep 2024

Dear Editors and Reviewers,

Thank you for your valuable comments and professional advice regarding our manuscript (ID: PONE-D-24-27892). We appreciate the insights provided, which have greatly contributed to enhancing the quality of our article. All authors have thoroughly discussed these comments, and we have carefully revised our manuscript to align with the requirements of your journal.

Reviewer comments are presented below in italics, with specific concerns numbered. Our responses are provided in normal font, and modifications made to the manuscript are indicated in red text. Please find our detailed responses to the reviewers as follows:

Reviewer #1:

1.In the introduction, the authors listed several current treatments for GERD, including PPIs, ARS and ECSF. Although they claimed that “the most widely performed form of ARS is laparoscopic antireflux surgery (LAS) ”, they focused solely on comparing the efficacy of various ECSF procedures with PPIs in patients with GERD. The rationale for excluding LAS from this comparison could be elaborated to provide further insights into their study design.

Response: We appreciate the thoughtful review and constructive feedback provided by the reviewers. LAS and ECSF are two very different sets of treatment modalities, and target different populations. LAS is a laparoscopic surgical procedure in patients with large hiatal hernia (>2cm) or severe reflux oesophagitis (LA-C grade or higher) by means of fundoplication of the gastric fundus. ECSF, on the other hand, is a group of endoscopic treatment modalities for patients with less severe reflux oesophagitis and oesophageal hiatal hernia (<2 cm), resulting in relatively fewer injuries and adverse effects. In summary, we considered that there are major differences between LAS and ECSF in terms of surgical methods and populations of recipients. In addition, studies directly comparing the efficacy of LAS with ECSF are scarce, not enabling META analyses to be performed. Last but not least, this study aimed at the effect of endoscopic treatment on GERD patients, so we did not put LAS in the selection.

2.The colors of the scatter plots in “Fig.6 The funnel plot for the included studies” should exhibit greater contrast, as the current colors make it difficult to distinguish between points.

Response: We sincerely appreciate your careful review and apologise for the oversight on the pictures. We have adjusted the scatter plots to a more contrasting colour. 

3.The manuscript does not clearly address why authors have chosen their outcome metrics. More literature about it should be cited.

Response: We sincerely appreciate the valuable comments. In this paper, GERD-Q score, PPIs discontinuation rate, AET, LES pressure, DeMeester score, adverse events, and patient satisfaction were chosen as outcome indicators.The GERD-Q score is a common clinical tool used for assessing reflux symptoms. 24-hour oesophageal pH monitoring and esophageal manometry are commonly used to clinically assess the efficiency of subjective and objective indicators. The AET, DeMeester score, is derived from 24-hour oesophageal pH monitoring and is commonly used to assess the severity of reflux. LES pressure is measured by oesophageal manometry to assess the presence of structural changes in the patient. Discontinuation rate of PPIs, adverse effects, patient satisfaction, on the other hand, are reliable indicators for assessing the patient's prognosis. We explain the choice of outcome indicators in the DISCUSSION section. And we have checked the literature carefully and added more references on and into the DISCUSSION section in the revised manuscript. Thanks again for your constructive advice.[Page 25, Line 314-316].

4.In the discussion, the authors claimed, “However, the effectiveness of each ECSF procedure for patients with reflux and the identification of an optimal choice remain unclear.” Consequently, the manuscript should address how current clinical treatment decisions are made within each ECSF procedure. Additionally, it should explore how the conclusions drawn from the manuscript can be applied to clinical decision-making.

Response: Thank you for your comments on this point. The selection of each procedure for ECSF in the current clinical practice is mainly based on the patient's own condition.In patients without severe reflux oesophagitis (LA class C or D) or oesophageal hiatal hernia greater than 2 cm, PECC is the preferred endoscopic treatment option, with fewer postoperative complications and lower medical costs. ARMI procedures are effective and safe for both PPI-refractory and PPI-dependent patients. Patients with extra-oesophageal GERD who have mild to moderate morphological damage to the gastro-oesophageal junction are the most desirable target population for ARMI. RF can be used in patients with failed LARS and can be used repeatedly as a remedial treatment after failed surgical procedures. In addition, RF is an option for patients with mild oesophagitis who do not have Barrett's oesophagus. Similarly, there may be a role for RF in patients with reflux hypersensitivity or functional heartburn. In clinical management, patients should complete various examinations, such as gastroscopy, reflux ultrasonography, oesophageal manometry, 24-hour PH monitoring, etc. According to the examination results combined with the patient's own symptoms, the most suitable treatment for the patient will be selected. According to the reviewer’s suggestions, we have added and refined the DISCUSSION section. Thank you again for your thoughtful comments.[Page 26, Line 338-341][Page 27, Line 352-357][Page 28, Line 371-375][Page 29, Line 381-385]

Reviewer #2:

1.Abstract: The authors report PECC is superior to RF with 2 numerical results (MD = -2.34; MD = 3.22), but then list three outcomes (GERD-Q score, LES pressure, and Serious adverse events). As readers will not be familiar with their approaches, it will not be obvious that the reported MD values apply to the first 2 outcomes. Please revise and clarify.

Response: We are very sorry for the lack of accuracy in our representation. Thank you very much for reading it carefully and pointing this out. We have amended the ABSTRACT section as follows: PECC was significantly superior to RF in lowering the patients’ postoperative GERD-Q scores(MD = -2.34, 95% confidence interval (CI):[ -3.02, -1.66]), augmentation of LES pressures(MD = 3.22, 95% CI: [1.21, 5.23]). Owing to limited data, we did not perform an NMA of adverse events after ECSF treatment. [Page 2, Line 33-37]

2.Abstract: In the next sentence, the authors report that ARMI was superior to "the other two," without specifying which "two" groups they're referring too, nor is it clear which outcomes the two reported results refer to. Please revise and clarify.

Response: Please accept our sincere apologies for the oversight in our writing. We value your input and we are grateful that you brought this issue to our attention. "the other two," referring to the PECC group and the RF group, we made the following changes in the ABSTRACT section: ARMI was preferable to PECC (MD = -2.87, 95% CI [-4.23, -1.51])and RF (MD = -1.12, 95% CI [-1.79, -0.54]) in reducing the AET percentage. We are committed to improving our writing and avoiding such mistakes in the future. Thank you for your understanding and support.[Page 2, Line 37-40]

3.Methods Section: Great job on providing a detailed explanation of methods.

Response: We are really glad to have your approval! Thank you again for your positive comments and valuable suggestions to improve the quality of our manuscripts.

4.Figures 3-5: These figures will be practically uninformative to all but the most statistically-trained readers. Please include (as appropriate) axis titles, legends, or footnotes to explain what's included in this Figures. In particular, where there are several panels in a Figure, please indicate which outcome or group each refers to. In Figure 4, please consider an alternative schema, as the dashed lines are difficult to distinguish.

Response: Thank you for your kind comments. We apologize for the poor thought that went into the addition of the legend and footnotes. At your suggestion we have modified the relevant charts. The titles of the modified charts have been highlighted in red and we hope that they will meet with your satisfaction.

Once again, we would like to express our sincere thanks to all the editors and reviewers. We hope this revised version will meet with the approval.

Yours sincerely,

Jun Zhang

The First Affiliated Hospital of Zhejiang Chinese Medical University

---

## [Decision Letter · Decision Letter 1]

17 Sep 2024

Network meta-analysis of the efficacy of endoscopic cardia peripheral tissue scar formation (ECSF) in the treatment of gastroesophageal reflux disease

PONE-D-24-27892R1

Dear Dr. Zhang,

We’re pleased to inform you that your manuscript has been judged scientifically suitable for publication and will be formally accepted for publication once it meets all outstanding technical requirements.

Kind regards,

Academic Editor

PLOS ONE

Additional Editor Comments (optional):

N/A

Reviewers' comments:

Reviewer's Responses to Questions

**Comments to the Author**

1. If the authors have adequately addressed your comments raised in a previous round of review and you feel that this manuscript is now acceptable for publication, you may indicate that here to bypass the “Comments to the Author” section, enter your conflict of interest statement in the “Confidential to Editor” section, and submit your "Accept" recommendation.

Reviewer #2: All comments have been addressed

2. Is the manuscript technically sound, and do the data support the conclusions?

Reviewer #2: (No Response)

3. Has the statistical analysis been performed appropriately and rigorously? 

Reviewer #2: (No Response)

4. Have the authors made all data underlying the findings in their manuscript fully available?

Reviewer #2: (No Response)

5. Is the manuscript presented in an intelligible fashion and written in standard English?

Reviewer #2: (No Response)

6. Review Comments to the Author

Reviewer #2: (No Response)

7. PLOS authors have the option to publish the peer review history of their article (what does this mean?). If published, this will include your full peer review and any attached files.

Reviewer #2: No

---

## [Editor Report · Acceptance letter]

13 Oct 2024

PONE-D-24-27892R1 

PLOS ONE

Dear Dr. Zhang, 

I'm pleased to inform you that your manuscript has been deemed suitable for publication in PLOS ONE. Congratulations! Your manuscript is now being handed over to our production team.

Kind regards, 

on behalf of

Dr. Jiangtao Gou 

Academic Editor

PLOS ONE